# Impact of MODIS sensor calibration updates on Greenland ice sheet surface reflectance and albedo trends

Kimberly A. Casey[1,2,*], Chris M. Polashenski[1,3], Justin Chen[4], Marco Tedesco[5,6]

[1] Thayer School of Engineering, Dartmouth College, Hanover, NH, 03755, USA

[2] Cryospheric Sciences Lab, NASA Goddard Space Flight Center, Greenbelt, MD, 20771, USA

[*] now at: [2] & Land Remote Sensing Program, U.S. Geological Survey, Reston, VA, 20192, USA

[3] Cold Regions Research and Engineering Laboratory, Alaska Projects Office, U.S. Army Corps of Engineers, Fairbanks, AK, 99709, USA

[4] Stanford University, Stanford, CA, 94305, USA

[5] Lamont-Doherty Earth Observatory, Columbia University, NY, 10964, USA

[6] NASA Goddard Institute for Space Studies, New York, NY, 10025, USA

*Correspondence to*: Kimberly A. Casey (kimberly.a.casey@nasa.gov)

## Abstract

We evaluate Greenland Ice Sheet (GrIS) surface reflectance and albedo trends using the newly released Collection 6 (C6) MODIS products over the period 2001-2016. We find that the correction of MODIS sensor degradation provided in the new C6 data products reduces the magnitude of the surface reflectance and albedo decline trends obtained from previous MODIS data (i.e. Collection 5, C5). Collection 5 and 6 data product analysis over GrIS is characterized by surface (i.e. wet vs. dry) and elevation (i.e. 500m-2000m, 2000m and greater) conditions over the summer season from June 1 – August 31. Notably, 20   the visible-wavelength declining reflectance trends identified in several bands of MODIS C5 data from previous studies are only slightly detected, at reduced magnitude in the C6 versions over the dry snow area. Declining albedo in the wet snow and ice area remains over the MODIS record in the C6 product, albeit at a lower magnitude than obtained using C5 data. Further analysis of C6 spectral reflectance trends show both reflectance increases and decreases in select bands and regions, suggesting that several competing processes are contributing to Greenland ice sheet albedo change. Investigators using MODIS data for 25   other ocean, atmosphere and/or land analyses are urged to consider similar re-examinations of trends previously established using C5 data.

## 1 Introduction

The Greenland ice sheet (GrIS) has experienced substantial mass loss during the past three decades resulting in sizeable contribution to sea level rise [Krabill et al., 2000; Fettweis, 2007; van den Broeke et al., 2009; Rignot et al., 2011; Velicogna 30   and Wahr, 2013; Enderlin et al., 2014]. Partitioned estimates of GrIS mass loss have shown that surface melt contributes substantially to annual ice loss [Box, 2013; van den Broeke et al., 2016]. Snow and ice surfaces become darker and have reduced visible-to-near-infrared reflectance and albedo due to the deposition of particulates, re-emergence of engrained

particulates, biological activity, snow grain metamorphosis and the presence of melt [LaChapelle, 1969; Warren and Wiscombe, 1980; Kohshima et al., 1993; Painter et al., 2001; Takeuchi, 2009; Hodson et al., 2017]. As surface melt is both a factor and result of surface darkening, the potential exists, for a variety of melt-albedo feedbacks to further enhance shortwave absorption and accelerate melt, increasing mass loss and sea level rise contributions [Wiscombe and Warren, 1980; Tedesco et al., 2015]. Moderate Resolution Imaging Spectroradiometer (MODIS) observations are among the best tools available to evaluate these albedo feedbacks. Several studies have used MODIS data to study the magnitude of ongoing albedo trends and assess their role in enhancing the impact of climate warming on the ice sheet [e.g. Tedesco et al., 2011; Box et al., 2012; Stroeve et al., 2013; Tedesco et al., 2016]. A challenge in using long term satellite records to detect change is maintaining consistent instrument performance. Recent literature has discussed MODIS Terra sensor calibration degradation and its impacts on apparent data trends [e.g. Franz et al., 2008; Wang et al., 2012; Lyapustin et al., 2014; Polashenski et al., 2015; Sayer et al., 2015]. Specifically related to GrIS trends, a recent work by Polashenski et al. [2015] indicated that uncorrected sensor degradation in MODIS Collection 5 (C5) data, particularly on the Terra platform, was contributing significantly to an apparent albedo declining trend over the GrIS and that the albedo trend in large areas of the ice sheet which do not experience melt (the dry snow area) may disappear once Collection 6 (C6) calibrations were applied. This study compares GrIS surface reflectance, snow and land albedo (Band 1-7) trends obtained from MODIS C5 with those obtained from the newly released MODIS C6 data to identify and differentiate the impact of sensor calibration drift from actual surface changes. We present these comparisons by spectral band and spatial filters and discuss the implications that C6 calibration enhancements have on our understanding of GrIS albedo decline and the mechanisms driving this decline. The identified Band 1-7, 459-2155 nm, and broadband albedo spatial and temporal patterns and trends provide insight toward the physical mechanisms likely dominating GrIS albedo decline, which is key to understanding the surface energy balance of the Greenland ice sheet.

## 2 Background

MODIS instruments operate onboard both the NASA Earth Observing System (EOS) Terra and Aqua satellites, and collect Earth observations in 36 spectral bands ranging from 0.4µm-14.4µm at spatial resolutions of 250m-1000m [Barnes et al., 1998]. The MODIS data record is now in excess of 17 years for Terra and 15 years for Aqua; both sensors are operating well past their 6-year design life. MODIS calibrations are updated periodically to reflect new understanding of instrument changes, with the entire data record reprocessed as a new 'Collection'. Significant revisions in the calibration approach were initiated in C6 [Toller et al., 2013], resulting in a relatively large adjustment to end products. Before launch, both MODIS sensors were calibrated via laboratory light sources. Because the optics of the sensor (including reflecting mirrors, electronics) were expected to degrade over time, MODIS sensor design included methods for post-launch onboard calibration. However, the onboard calibration did not sufficiently account for degradation of the calibrators due to the large degradation of the solar diffuser [detailed in Lyapustin et al., 2014]. This led to a long-term drift in calibration, most pronounced on the Terra sensor, largest in the blue band (B3), and decreasing with increasing wavelength [Xiong and Barnes, 2006].

The discovery of systemic non-physical trends in MODIS Terra products by science users [e.g. Kwiatkowska et al., 2008; Levy et al., 2010; Wang et al., 2012] motivated research into independent trend characterization for C6 calibration. Analysis of observations of remote desert targets that were assumed to be nearly invariant over the long term were used to constrain the long-term drift of the observations. These so-called pseudo-invariant targets such as deserts, deep convective clouds and high

elevation ice sheet investigations led to a new vicarious approach for calibration, which relies on collection of pseudo-invariant Earth site data continually to provide a reference over time. The approach permits calibration at multiple angles of incidence (AOI) on the challenging-to-characterize scan mirror, rather than two angles of incidence available through the onboard solar diffuser and moon observations through the space view port [Sun et al., 2012; Lyapustin et al., 2014]. The resulting calibration is of lower precision than a well-characterized mirror calibrated on a lunar standard, but provides significant improvement to

the C5 data [Lyapustin et al., 2014]. Long-term degradation of the solar diffuser stability monitor and detectors will continue to be evaluated and addressed as the MODIS record proceeds [Toller et al., 2013].

The impact of C5 to C6 updates on higher-level MODIS products cannot be represented by a simple offset because the magnitude of calibration revision is dependent on mirror side, AOI and time. The correction reaches several percent (in absolute reflectance values) in the worst cases (i.e. B3, near the end of the C5 record, roughly 2013 to 2015). Lyapustin et al.

[2014] report C5 to C6 adjustment of B3 top-of-atmosphere (TOA) reflectance reaches approximately 0.02 (from 0.225 to 0.245) at the Libya 4 pseudo-invariant site by Terra mission day 5000. Residual error after C6 calibrations (when compared to pseudo-invariant desert sites) is found to be on the order of several tenths of a 1% in TOA reflectance [Lyapustin et al., 2014]. The residual error is now within the stated accuracy of the products, but may still be large enough to impact end user's scientific results, particularly in products derived from band ratios such as aerosol or vegetation indices. Further possible

improvements to sensor calibration have been discussed [Meister et al., 2014; Xiong et al., 2015] including those that address polarization correction on the Terra sensor [Lyapustin et al., 2014]. A thorough description of the C5 calibration degradation can be found in Lyapustin et al. [2014], and further details on C6 sensor characterization can be found in Toller et al. [2013] and at the MODIS Characterization Support Team (MCST) website http://mcst.gsfc.nasa.gov/calibration/information. MODIS C6 updates included not only sensor calibration algorithms applied to Level 1 data, but also updates to algorithms used to

derive higher level data products (including products used in this study), as well as the aerosol retrieval and correction algorithms, the cloud and cloud shadow detection algorithms, and quality assurance bands [Levy et al., 2013; Platnick et al., 2015]. Detailed documentation of modifications to the products used in this paper can be found in the MOD09 user guide at http://modis-sr.ltdri.org/guide/MOD09_UserGuide_v1.4.pdf, the MOD10 user guide at http://modis-snow-ice.gsfc.nasa.gov/uploads/C6_MODIS_Snow_User_Guide.pdf and the MCD43 documentation at

https://www.umb.edu/spectralmass/terra_aqua_modis/v006. Our analysis investigates Terra and Aqua records from three product types, namely MOD09/MYD09, MOD10/MYD10, and MCD43 for the entire summer data record, though does not directly compare MODIS C5 and C6 data with ground data. Several MODIS cryospheric calibration and validation investigations, some of which include in situ data, have been performed, such as Stroeve et al., [2005]; Moody et al., [2007]; Hall et al., [2008]; Stroeve et al., [2013]; Alexander et al., [2014]; Wright et al., [2014]; Polashenski et al., [2015]; Zhan and

Davies, [2016]. Recently, Box and others [2017] found Terra MOD10A1 albedo substantially improves in relative accuracy from C5 to C6, agreeing with GC-Net and PROMICE station data, from mid-May through August for the majority of GrIS south of 80°N within 0.04 (unitless albedo from 0-1, see Box et al., 2017, Figure 5b). Accuracy of in situ ice sheet automated weather station measurements remains challenging due to limitations of unattended stations and interference of factors including ice riming, high wind speeds, low temperatures, ablation, tilt and ice flow [van den Broeke et al., 2004; van As, 2011]. Considerable biases have been reported in GrIS automated weather station data [Stroeve et al. 2006; Stroeve et al., 2013; Wang et al., 2016; Ryan et al., 2017].

## 3 Methods

### 3.1 Processing MODIS Data

We analysed the MODIS Terra and Aqua 8-day surface reflectance products (MOD09A1/MYD09A1) [Vermote, 2007; Vermote, 2015], Terra and Aqua daily snow cover and broadband albedo products (MOD10A1/MYD10A1) [Hall et al., 2006; Hall and Riggs, 2016], and combined Terra and Aqua platform daily land surface albedo products (MCD43A3) [Schaaf and Wang, 2015] for both C5 and C6 collections over Greenland. The MOD09A1 and MYD09A1 land surface reflectance products, from the Terra and Aqua platforms, respectively, provide surface reflectance in bands 1-7, corresponding to a wavelength range of (459 – 2155 nm) (Table 1). Data is provided as an 8-day product, which contains the best available Level 2 gridded observation during the 8-day period, based on observation coverage, view angle, clouds and aerosol loading. The MOD10A1/MYD10A1 Terra/Aqua daily snow cover products include daily broadband snow albedo and quality assurance observations, which are the focus of our processing, in addition to a snow cover index calculated from Level 1 radiance data. The MCD43A3 land surface albedo product provides both direct hemispherical reflectance (black sky albedo, BSA) and bihemispherical reflectance (white sky albedo, WSA) for bands 1-7 from a bidirectional reflectance distribution function (BRDF) inversion of all available observations during the 16-day moving window centered on the date of interest. Data are provided every 8 days in C5 and daily in C6.

Because the high latitude and low solar zenith angles over the GrIS are at the extreme of MODIS capabilities, we filtered the data to ensure use of only the highest quality retrievals for our analysis. MOD09A1/MYD09A1 (M*D09A1) data were filtered by using the band quality assurance layer where all four band quality flags were set to 0 for each band, and solar zenith angle observations were below 70 degrees. MOD10A1/MYD10A1 (M*D10A1) daily snow cover data were filtered using methods similar to those in Box et al., [2012], where values outside M*D10A1 scientific data set albedo values are excluded and a filter removes pixels whose albedo is more than 2 standard deviations from the 11-day running median or which differ from the 11-day running mean by more than 0.04 (albedo is a dimensionless number on a scale of 0.0-1.0). Processing MCD43A3 datasets was carried out following methods described in Stroeve et al. [2013] section 2.1. We use only data with highest quality (inversion flag set to 0 in the MCD43A2 product), which represents data derived from time periods where sufficient cloud-

free, high quality observations are available for full BRDF inversion. We did not use data derived with the backup algorithm, even though evidence suggests it performs almost as well [Stroeve et al., 2013]. During the summer season chosen for the processing timeframe (Jun 1 – Aug 31), the majority of data is acquired near solar noon, ensuring many high-quality inversions in the MCD43A3 data.

After quality filtering, data is processed to produce an annual mean albedo for dry and wet snow and ice areas of the ice sheet with methods of Polashenski et al., [2015]. Wet snow and ice are GrIS areas that have experienced melt at any point during the current year prior to the date of interest. Dry snow is snow which has not at any point experienced melt. We do not return wet snow and ice to the dry snow category after it experiences melt until the following year when we are certain the melt surface has been buried, due to the residual impacts on albedo caused by melt occurrence. An elevation mask is applied using

a GrIS digital elevation model (DEM) [Howat et al., 2014] to group ice sheet areas at two different elevation bands (dry snow = snow above 2000m, wet snow and ice = snow and ice within the elevations of 500 to 2000m) (Figure 1). To these two elevation areas, high elevation and low elevation, we apply a melt mask generated from the regional climate model Modèle Atmosphérique Régional (MAR) [Tedesco, 2014] to exclude pixels that did not match the predominant melt condition. MAR indicated dry and wet snow and ice pixels were defined as those with no simulated melt occurring at any time during the

summer (dry), and those showing one or more melt events (wet). The use of a dual filter, based on elevation and melt state, ensures that the areas discussed represent iconic surface types without contamination (e.g. for the dry snow area, the elevation cut off ensures non-melting bare ice at the margins of the ice sheet does not contribute). From the mosaicked, filtered and masked data, the average of the daily mean of all remaining pixels from June 1 – August 31 is calculated for each year of the record. Though the time interval differs from the May 15 – July 15 time interval chosen by Polashenski et al. [2015] to match

the time when solar elevation angle is highest and albedo is most important to the GrIS, it is better aligned with prior studies and still captures the key behavior of the high insolation period. We also conducted all the same analysis for May 15 – July 15 (not shown) and found very similar trend revisions and overall behavior of ice sheet albedo. Linear trends are calculated for display in map form using a least squares linear regression to the data at each pixel location.

  All MODIS data product tiles were downloaded from the NASA Land Processes Distributed Active Archive Center (LP

DAAC) and National Snow & Ice Data Center (NSIDC). Data tiles were mosaicked and resampled via nearest neighbor method to polar stereographic projection using the MODIS Reprojection tool. The MODIS Reprojection Tool utilized to mosaic and resample MODIS tile data can be found at https://lpdaac.usgs.gov/tools/modis_reprojection_tool and the user guide at https://lpdaac.usgs.gov/sites/default/files/public/mrt41_usermanual_032811.pdf.

# 4 Results

## 4.1 Annual Average Summer M*D09A1 Surface Reflectance and M*D10A1 Broadband Albedo

Annual average summer (June 1 – August 31) GrIS surface reflectance for dry snow and wet snow and ice areas, from both Terra and Aqua data is presented in Figure 2 and 3, respectively, as derived from the M*D09A1 and M*D10A1 C5 and C6 products. The discrepancy between the two data collections is indicated by the difference between dashed (C5) and solid (C6) lines of the same color. The adjustment from C5 to C6 is significantly greater for Terra than Aqua and greatest over the GrIS in the shortest wavelength bands, consistent with the sensor calibration degradation reported by Lyapustin et al. [2014]. C6 reduces the discrepancy between Terra and Aqua data appreciably. Trends of the plotted data are quantified by linear regression in Tables 2 and 3 along with their statistical significance. Significant declining trends found in C5 Terra dry and wet snow and ice data with magnitude exceeding 0.01/decade are no longer present in C6 data. Very small dry snow trends remain in C6 data, though not of strong statistical significance. Thus, these likely do not represent real changes on the surface. The C6 trend magnitude over the dry snow area is near the calibration drift of several tenths of percent and the trends show an incoherent pattern of albedo change. Specifically, B3, 459-479 nm increases slightly, while B1, 620-670 nm decreases. This could not be produced by expected physical mechanisms, for example, absorbing impurity concentration or grain size changes, which would both be expected to shift B1 and B3 in the same direction [see Warren and Wiscombe, 1980; Wiscombe and Warren, 1980, respectively]. In wet snow and ice, significant trends seen in C5 Terra albedo nearly disappear in C6. Marginally, non-significant trends in wet snow and ice albedo, remain across C6 visible bands, at magnitudes approximately one third to one half those of C5 data. We note that higher interannual variability (noise) in the wet snow and bare ice area limits trend significance at the $p \leq 0.05$ level, even though absolute magnitude of trends (signal) is larger than in the dry snow area. The coherence across visible bands of marginally non-significant trends, however, (nearly all bands decline for both sensors) and magnitude exceeding sensor accuracy suggests a physically real trend is likely, if not proven statistically by the satellite data products.

## 4.2 Annual Average Summer MCD43A3 Albedo

Annual average summer (Jun 1 – Aug 31) GrIS C5 and C6 MCD43A3 land surface direct hemispherical reflectance are presented in Figure 4 for dry and wet snow and ice, showing results expected from a combination of Terra and Aqua data. Dashed lines represent C5 data, while solid lines represent C6. Revisions across the duration of the MODIS record are apparent between the MCD43A3 C5 and C6 data in both dry and wet snow/ice areas. Similar to M*D09A1 and M*D10A1 products, the MCD43A3 revisions result in a considerable decrease in trend magnitude. Statistically significant dry snow direct hemispherical reflectance declines, which had been apparent in C5 data, are reduced to magnitudes at or under 0.01 per decade in C6. MCD43A3 direct hemispherical and bihemispherical reflectance trends (Table 3) remain significant or near-significant for B1 and B4, with incoherent patterns for other visible bands (B2, B3). As discussed in section 4.1, the spectral pattern, with

conflicting trends in B2 and B3 indicates these changes are likely not linked to physical processes. Wet snow and ice areas still exhibit coherent declining albedo trends after C6 revisions across B1 through B7, albeit of slightly lower magnitude than in C5. Like C5, individual band wet snow and ice trends have low (B2, B3) to marginal (B1) statistical significance due to large interannual variability in the wet snow and ice albedo (Figure 4). Coherence across bands (as presented in Table 3 BSA

C6 Wet Snow/Ice and WSA C6 Wet Snow/Ice trend columns), however, increases confidence in the trends. Impacts of C5 to C6 revision are very similar on WSA and BSA.

## 4.3 Spatial Pattern of Albedo Trend

Maps of the decadal M*D10A1 broadband snow albedo trend over the entire MODIS record are shown in Figure 5 for C5

(a,d) and C6 (b,c,e,f) data from both Terra and Aqua sensors. Both sensors and both collections show similar spatial patterns, with greatest albedo declines at low elevations of the ice sheet (Figure 1), particularly on the western and southeastern margins. A statistically significant, ice sheet-wide decline albedo trend in C5 Terra data is largely absent in C5 Aqua sensor data. The discrepancy between C5 Terra and Aqua data is not spatially dependent. C6 revisions change Terra trends upward by approximately 0.03/decade across the ice sheet. Aqua revisions are smaller, but also result in trends that are less negative in

C6 than C5. Using revised C6 data, we mask the areas that have negligible trend (-0.01 to +0.01/decade) in Figure 5 c,f. The magnitude of a 'negligible' trend was determined by considering the errors of several tenths of a percent trend per decade remaining in C6 data collected to that obtained over pseudo-invariant desert sites [Lyapustin et al., 2014]. Trends below this value must be considered below detection limit. The region of negligible trends covers nearly all of the dry snow and large portions of the upper reaches of the wet snow and ice, indicating that even in some areas where it is well known that melt

frequency is increasing [e.g. Fettweis et al., 2011; Box et al., 2012; Fausto et al., 2016], albedo trends are small enough to be challenging to confirm over the duration of the MODIS record. Trends with magnitude greater than sensor calibration limits of ~0.01/decade are almost all negative. Significant areas of the southern ice sheet exhibit trends near -0.01/decade and a narrow band of substantial albedo decline, reaching -0.04/decade, exists in areas around the periphery of the ice sheet. Based on MAR analysis, the positive albedo trends over northeastern Greenland are likely associated with a shift from no trend in

accumulation to a trend of accumulation increasing (by 35 Gt/yr), starting in 2013. Direct hemispherical reflectance trends for select visible and near infrared bands of MCD43A3 C6 data are presented in Figure 6. Both positive and negative trends of statistical significance exist in all bands around the periphery of the ice sheet. These are discussed in detail below. Band 1 and Band 3 data show spatially uniform trends that are generally under sensor calibration accuracy of ~0.01/decade across the interior of the ice sheet. Band 1 red visible light trends are positive and Band 3 blue visible light trends are negative, a behavior

inconsistent with albedo change caused by either light absorbing impurity deposition, or changes in surface grain properties, strongly suggesting these changes are not physically real. NIR bands 2 and 5 show trends in the upper elevations of the ice sheet that regionally exceed sensor calibration accuracy and significance, with notable spatial patterns. Positive albedo trends in these NIR bands dominate northeast Greenland, while negative trends cover most of the remainder of the ice sheet. Band

5, in the near infrared, is highly sensitive to grain size impacts. Figure 6d shows the opposing trends of albedo increasing primarily in the northeast and decreasing in the west and southern periphery. Figure 4, B5 shows modest interannual variability in the dry snow and more variability in the wet snow and ice areas, suggesting wet snow and ice albedo is worth investigating on seasonal and/or annual scales.

## 5 Discussion

### 5.1 Impact of C6 Revision on Scientific Investigation of GrIS

The MODIS C5 to C6 revisions, and new surface reflectance and albedo trends over GrIS, have substantial ramifications for research seeking to evaluate the cause of enhanced surface melt, and hence mass loss, from the GrIS. Over the dry snow area, C5 Terra data and combined sensor data indicated a decadal trend of declining reflectance at a rate of up to several percent per decade, with high statistical significance (see Tables 2, 3, Figure 5a). The albedo decline was greatest in short wavelength visible bands – a spectral signature consistent with enhanced dust deposition on the ice sheet [Dumont et al., 2014]. The data, therefore, suggested that a snow albedo feedback initiated by dry snow processes, resulted in increased GrIS melt. C6 revisions, particularly to short wavelength bands of Terra data, appreciably reduce these trends. C6 exhibits no statistically significant trends in visible nor NIR wavelength band surface reflectance from either sensor over the aggregated dry snow area exceeding sensor calibration accuracy of ~0.01/decade. Statistical significance of trends in MCD43 B4 C6 data is misleading. These trends, of a few tenths of a percent per decade, are within the range of residual calibration error and their statistical significance may in fact reflect the ease with which a calibration error can be detected as a significant trend when superimposed on the relatively constant albedo of dry snow. Care in examination of the spatial and spectral variability and coherency between these trends (Figures 5, 6) is recommended for future regional, in situ and/or process studies. Trends in C6 dry snow visible albedo (B1 and B3) are suspiciously consistent across the ice sheet, and opposite trends between these bands are inconsistent with expected mechanisms of albedo change. We conclude that dry snow visible band albedo is stable within MODIS' capabilities. This stable visible wavelength albedo of the GrIS dry snow area supports conclusions in Polashenski et al. [2015], who found no in situ evidence of continual enhanced black carbon or dust deposition to support C5 trends. To note, forest fire events in North America and Asia have resulted in black carbon deposition to GrIS [e.g. Zennaro et al., 2014; Thomas et al., 2017] and such events have been predicted to increase during periods of drought in a warming climate [Soja et al., 2007; Flannigan et al., 2013]. Deposition of black carbon and other absorbing impurities in the dry snow area is often buried by new snowfall, however these impurities often have stronger influence in reducing reflectance and albedo in wet snow and ice areas. In contrast to B1 and B3 decadal trends which are nearly stable in aggregate, NIR wavelength albedos (B2 and B5) show significant regional trends. Positive trends in NE Greenland offset negative trends across much of the remainder of the ice sheet in the aggregate dry snow data. The regional trends are statistically significant over large areas and exceed the expected magnitude of remaining calibration errors. The trends appear to indicate changing snow grain metamorphism. We speculate,

based on the spectral change, that enhanced snowfall in the dry areas of NE Greenland is causing more rapid surface burial and lower age of surface grains, while increased temperatures and occasional melt are increasing grain size on southern and western areas of GrIS, where deposition was already relatively frequent. We find that the pattern does not appear to coincide well with the area of enhanced melt, as detected by passive microwave products [Tedesco et al., 2014] over the MODIS era, indicating that this indeed appears to be more related to dry snow grain metamorphism and snowfall frequency than melt (see Figure 7).

In wet snow and ice areas, the annual average broadband albedo and visible wavelength reflectance shown for all products exhibits a downward trend for C5 and C6 data. The magnitude of these trends is reduced by approximately one half from C5 to C6 for MOD09A1 and by a small amount in MYD09A1 and MCD43A3. Statistical significance is harder to establish in the wet snow and ice areas due to much higher interannual variability (wet snow and ice, Figures 2, 3, and 4). The C5 to C6 revisions to wet snow and ice albedo trends impact our understanding of GrIS surface energy and mass budgets and suggest a smaller role for albedo feedback in driving Greenland mass loss than previously indicated. The C5 to C6 revisions do not, however, demand a change in conclusions about reflectance and albedo declining trend existence in the wet snow and ice areas. Wet snow and ice areas on GrIS, in aggregate, still show coherent albedo decline across the nearly all visible and NIR wavelength bands from both sensors in C6 data (Tables 2, 3). Reduction in reflectance and albedo of this magnitude (several tenths) has been shown to result in radiative forcing of several tens of W m$^{-2}$ [e.g. 0.4 broadband albedo decline from dust on snow resulting in 80 W m$^{-2}$ radiative forcing, Painter et al., 2007; 0.3 reduction in broadband visible albedo from black carbon and impurities on snow resulting in 70 W m$^{-2}$ radiative forcing, Casey et al., 2017]. The MODIS C6 reflectance and albedo data product results provide strong supporting evidence that enhanced melt processes (including melt induced snow microstructure changes and melt induced light absorbing impurity accumulation changes) are creating an albedo feedback on the GrIS periphery.

The cause of these changes is important. We examined trends regionally along the margins of the ice sheet (Figures 5 and 6), and find that trends can be supported by reasonable inferences about regional differences in snowfall frequency, melt duration and surface exposure of light absorbing impurities on albedo control. In west Greenland, from Humbolt glacier to Jakobshavn glacier, substantial trends in NIR (B2, B5) albedo across a wide elevation range indicate increased presence of melting conditions. Trends in visible albedo are spatially correlated, and most dominant at lower elevations where melt accumulation of impurities, exposure of bare ice and algal growth occurs. South of Jakobshavn on the Western margin of the ice sheet, B1 and B3 visible albedo are sharply declining but are accompanied by rising B5 NIR albedo. We speculate that this signature could be evidence for substantial surface exposure and accumulation of mineral impurities in this region of GrIS. The spectra of most minerals exhibit higher NIR reflectance than bare ice, leading to the potential for a trend toward increased NIR albedo while visible albedo drops with high mineral content [Adams and Filice, 1967; Painter et al., 2003; Bøggild et al., 2010; Casey et al., 2012; Tedesco et al., 2013]. The SE GrIS margin shows decreasing NIR albedo, and only isolated change in visible wavelengths at the lowest elevations. We interpret this similarly to NW Greenland, only with much higher accumulation in this region. NIR impacts of increased wet snow and ice presence dominate. Higher accumulation drives the equilibrium line

to a lower elevation and melt accumulation of light absorbing impurities plays a substantial role in lowering visible albedo below this elevation. Continuing around the periphery of the GrIS north, the remainder of the ice sheet margin from approximately Scoresbysund to Humbolt Glacier, we see positive trends in NIR albedo and no significant trends in short wavelength visible albedo. In this region of low annual accumulation and long surface exposure times, this signature appears

to indicate increasing snowfall, which is confirmed by our separate preliminary analysis of MAR data in northeastern Greenland showing an increase in accumulation patterns starting in 2013 as well as GrIS surface mass balance climate model results [e.g. Noël et al., 2015]. A very small addition of snowfall would cause more rapid surface burial and result in lower age of surface grains and higher NIR albedo. Ultimately, each of these interpretations only clarifies what physical mechanisms would be consistent with the spectral signature changes observed. These hypotheses should be considered provisional and

tested by in situ observation of snow and ice properties, which may be guided by satellite identified signals.

## 5.2 Future Use of MODIS Data

The MODIS record is a powerful tool for assessing surface reflectance and albedo changes remotely. Our results indicate that future investigations should use the latest data recalibration (currently C6) data, and investigators should be aware of the

limitations of the sensors, which we here attempt to restate plainly for the community's benefit:

1.  Absolute trends in reflectance/albedo on the order of 0.01/decade are near the limits of the sensor calibration accuracy. Though statistically significant albedo trends of 0.01 or less TOA reflectance may exist over some surfaces with particularly stable albedo, trends at this level should be considered provisional and evaluated with great care, as they might not reflect actual physical processes.

2.  Calibration drift is band dependent. Small band-dependent calibration degradations can by magnified in band ratio products, such as those used to detect dust mineralogy or aerosols, indicating spurious trends.

3.  Data limitations are greatest in recently collected data. Vicarious C6 calibration, based on pseudo-invariant Earth sites, may not fully capture emerging trends in sensor degradation. Increasing Terra-Aqua discrepancies appear since 2014 in C6 data.

4.  When the sensors disagree in ways not explained by overpass time, MODIS Aqua is likely to provide more stable data as Terra's calibration is expected to continue to degrade in ways that will make it challenging to characterize. Since C6 calibration is now vicarious (based on observation of pseudo-invariant desert sites), it is likely that emerging trends in sensor behaviour will take some time to manifest in a statistically significant way for calibration revisions.

## 6 Conclusions

MODIS C6 calibration revisions result in substantial modification of decadal albedo trends on the GrIS reported by prior authors based on C5 data [e.g. Box et al., 2012; Stroeve et al., 2013; He et al., 2013; Polashenski et al., 2015]. Declining C5 surface reflectance trends that were particularly pronounced in Terra's shortest wavelength bands, are smaller or absent in C6

data. MODIS C6 surface reflectance and albedo data over dry snow areas of the GrIS features mostly small, non-statistically significant trends in visible and NIR wavelengths. (The exception is marginally statistically significant B1 decline in BSA and WSA dry snow albedo, and statistically significant B4 decline in BSA and WSA dry snow albedo.) These findings are consistent with the recent study of Polashenski et al. [2015] who suggested the dry snow albedo decline in C5 data would

disappear in C6 after finding no enhancement in light absorbing impurity concentrations on the interior Greenland ice sheet. The declining trends in wet snow and ice surface reflectance and albedo, independently supported by evidence of increased melt activity [Nghiem et al., 2012; Fausto et al., 2016b], remain statistically significant in C6 data, though at lower magnitude. An examination of spatial, wavelength-specific variability in C6 albedo trends indicates several interesting attributes of GrIS albedo decline that may motivate future work to better understand the mechanisms controlling albedo feedbacks on the ice

sheet. At higher elevations, patterns of NIR albedo change, including increasing reflectance in NE Greenland and declining reflectance in Southern and Western Greenland, highlight possible regional changes in metamorphism, precipitation and surface constituents. In the ablation zone, ratios of visible and NIR albedo trends suggest differences in snowfall frequency, melt duration and surface exposure of light absorbing impurities control recent albedo trends – the net impact of these mechanisms being complex, and likely dependent on the location and the seasonal timeframe chosen. Though the majority of

albedo reduction occurs on the GrIS in melt impacted areas, these results may support a crucial role for snow grain metamorphism in initiating (or preventing) feedbacks in dry snow.

Melt-related albedo reductions continue to have potential to trigger significant ice-albedo feedbacks and accelerate melt and surface mass loss, and, as a result, melt initiation remains a critical process. Our findings, particularly those illustrating the regional complexity in spectral albedo trends, highlight the need for future work on GrIS albedo to define and differentiate the

role of processes that control albedo decline.

The implications of this study extend beyond Greenland. The substantial revision from C5 to C6 MODIS products impacts a broad array of investigations. Conclusions based upon trends from C5 data, particularly shorter wavelength band Terra data, should be re-examined for robustness with C6 products. Future investigators should also note the limitations of MODIS products. Investigators should use great caution in evaluating trends of ~0.01/decade or smaller, and note that C6 corrections

may not fully capture recent and emerging trends in sensor degradation, particularly on the challenging to characterize Terra sensor.

**Acknowledgments**

We acknowledge funding from NSF Grants ARC-1204145, 1304807; NASA Grants NNX14AE72G, NNX14AD98G,

NNX16AO75G. We thank NASA EOS, LP DAAC and NSIDC for providing MODIS data and C. Schaaf and Q. Sun for providing an executable file to grid recently released ungridded MCD43 C5 data. We thank Dr. Marie Dumont for her service as editor, as well as Dr. Jason Box and an anonymous reviewer for constructive comments which improved this manuscript.

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

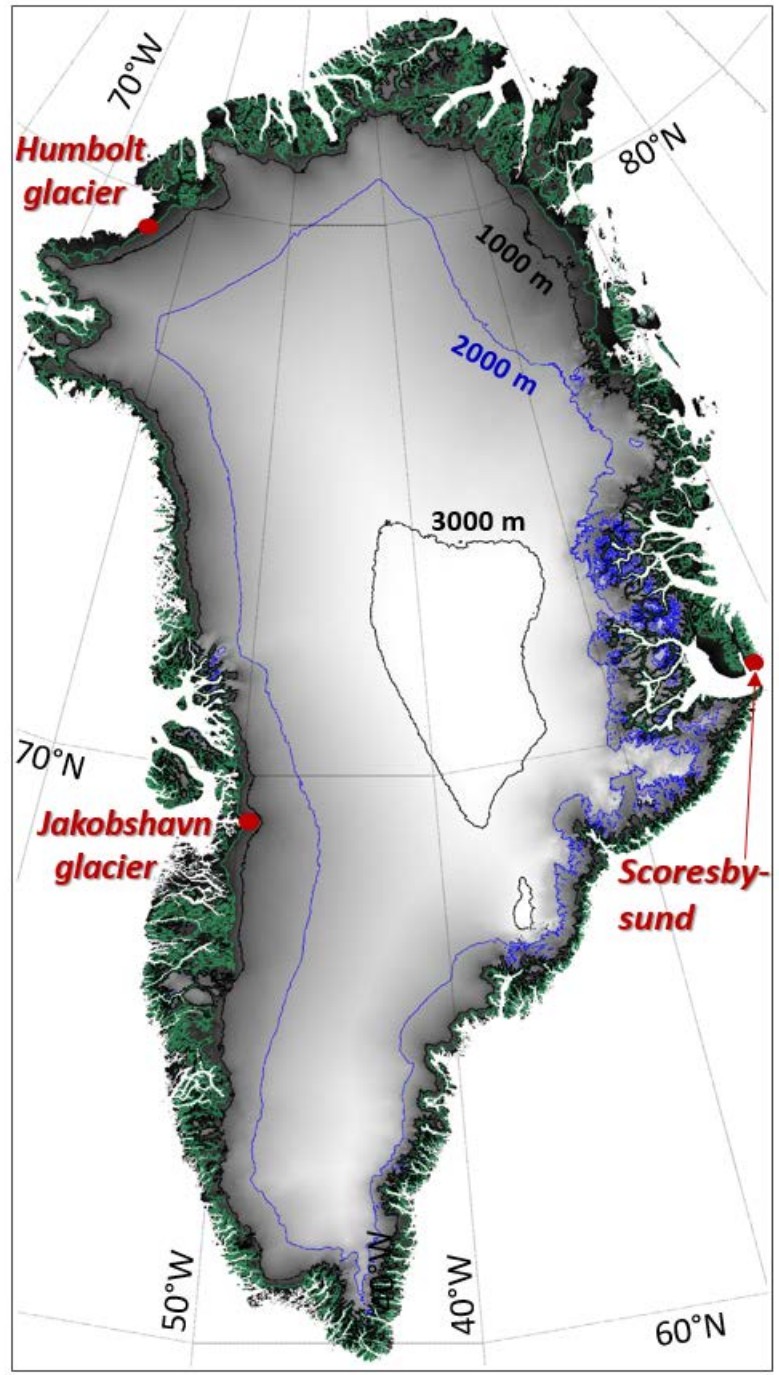

**Figure 1: Location and topographic map of the Greenland ice sheet with 500 m (green), 1000 m (black), 2000 m (blue) and 3000 m (black) contour lines overlaid. Greenland surface elevation from Howat et al [2014] is displayed where darker greys indicate lower elevation (minimum at sea level, 0m) and brighter greys and white indicate higher elevation (maximum at 3500m).**

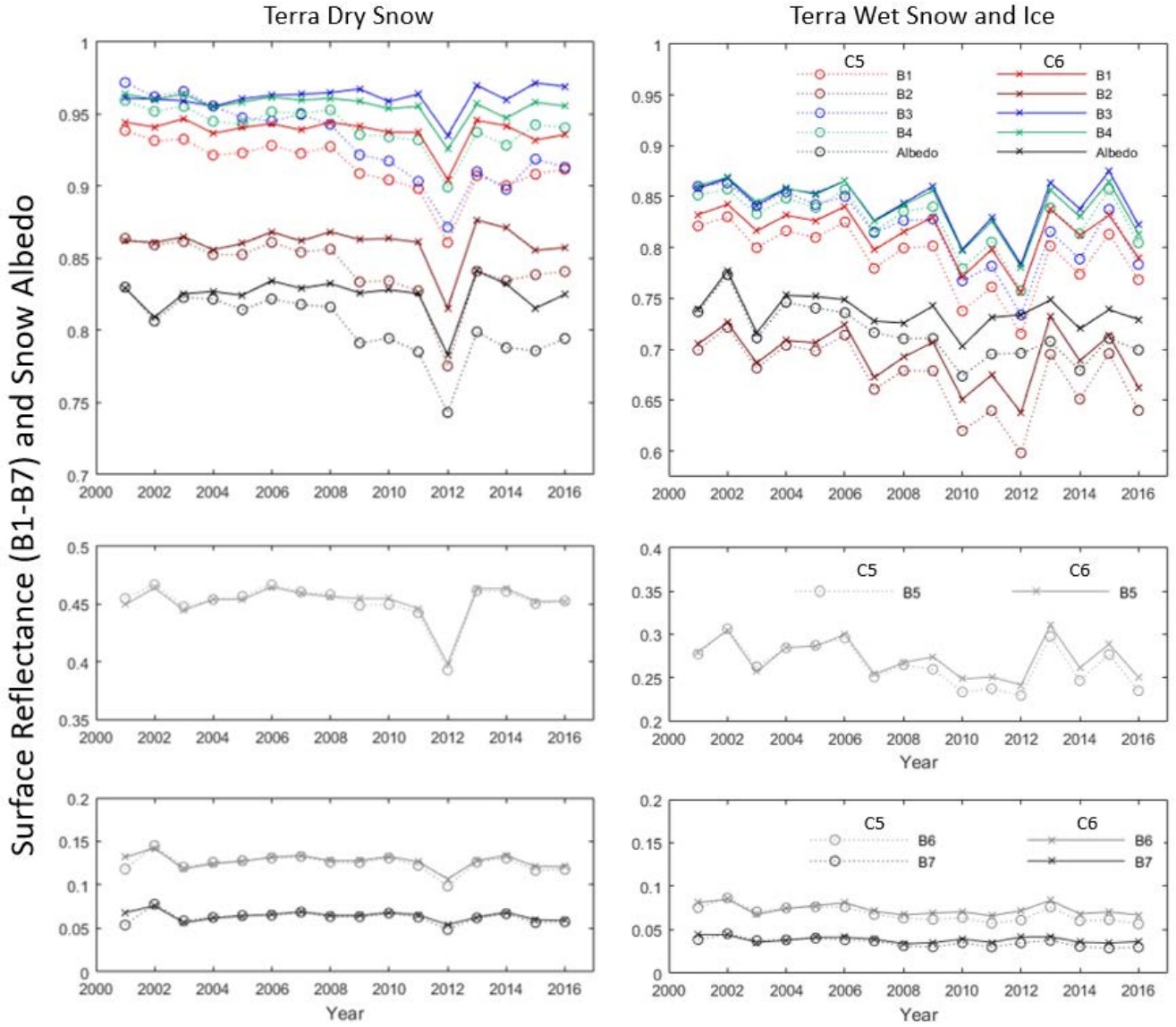

**Figure 2: Terra MODIS GrIS C5 (dashed) and C6 (solid) average annual Jun 1 - Aug 31 MOD09A1 surface reflectance (B1-B7) and MOD10A1 broadband snow albedo, denoted 'albedo' for dry (left) and wet snow and ice (right). (Note the y-axis scale is different for dry vs. wet snow and ice for B1-4, Albedo and B5).**

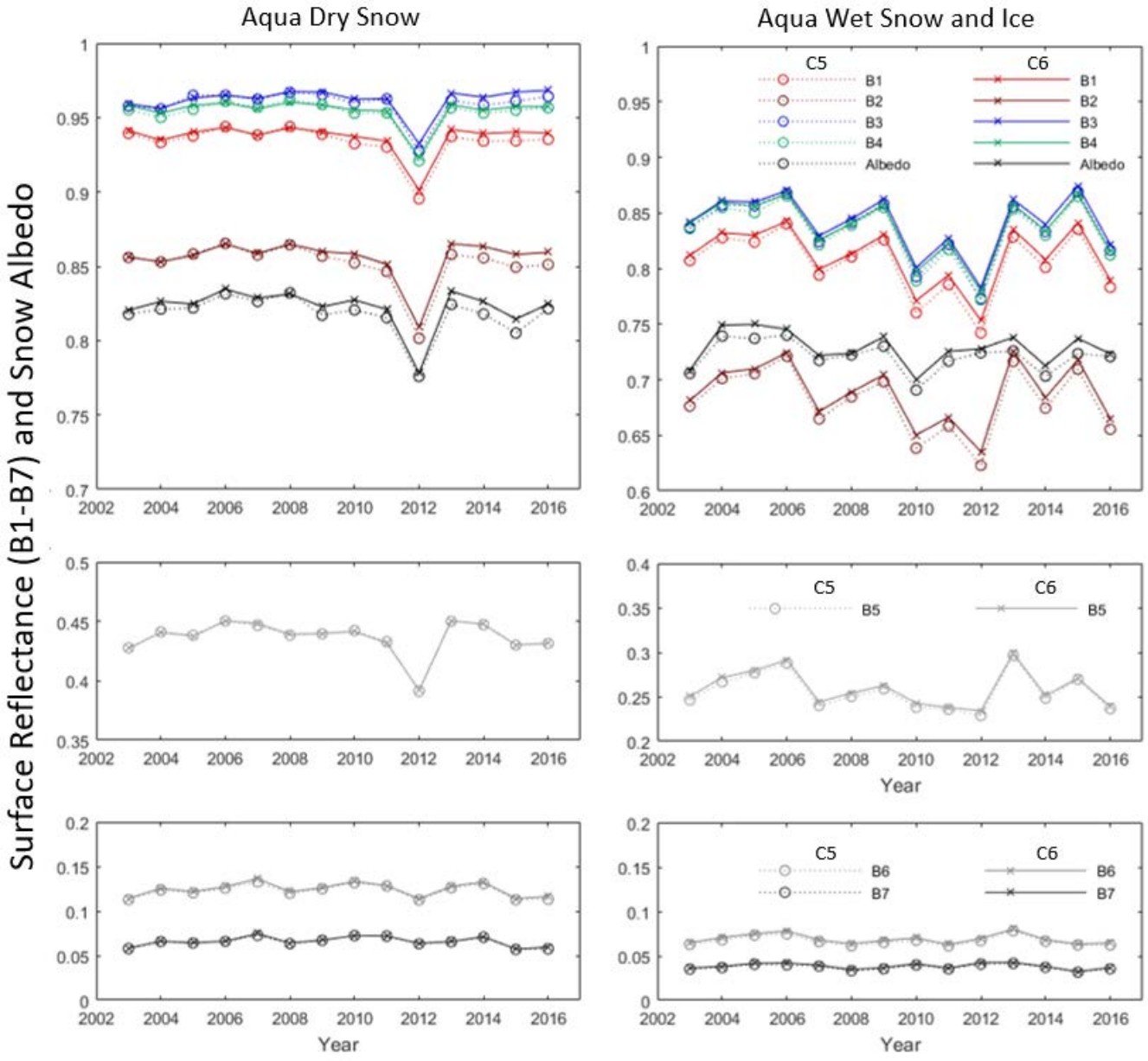

**Figure 3: Aqua MODIS GrIS C5 (dashed) and C6 (solid) average annual Jun 1 - Aug 31 MYD09A1 surface reflectance (B1-B7) and MYD10A1 (broadband snow albedo, denoted 'albedo') for dry (left) and wet snow and ice (right). (Note the y-axis scale is different for dry vs. wet snow and ice for B1-4, Albedo and B5).**

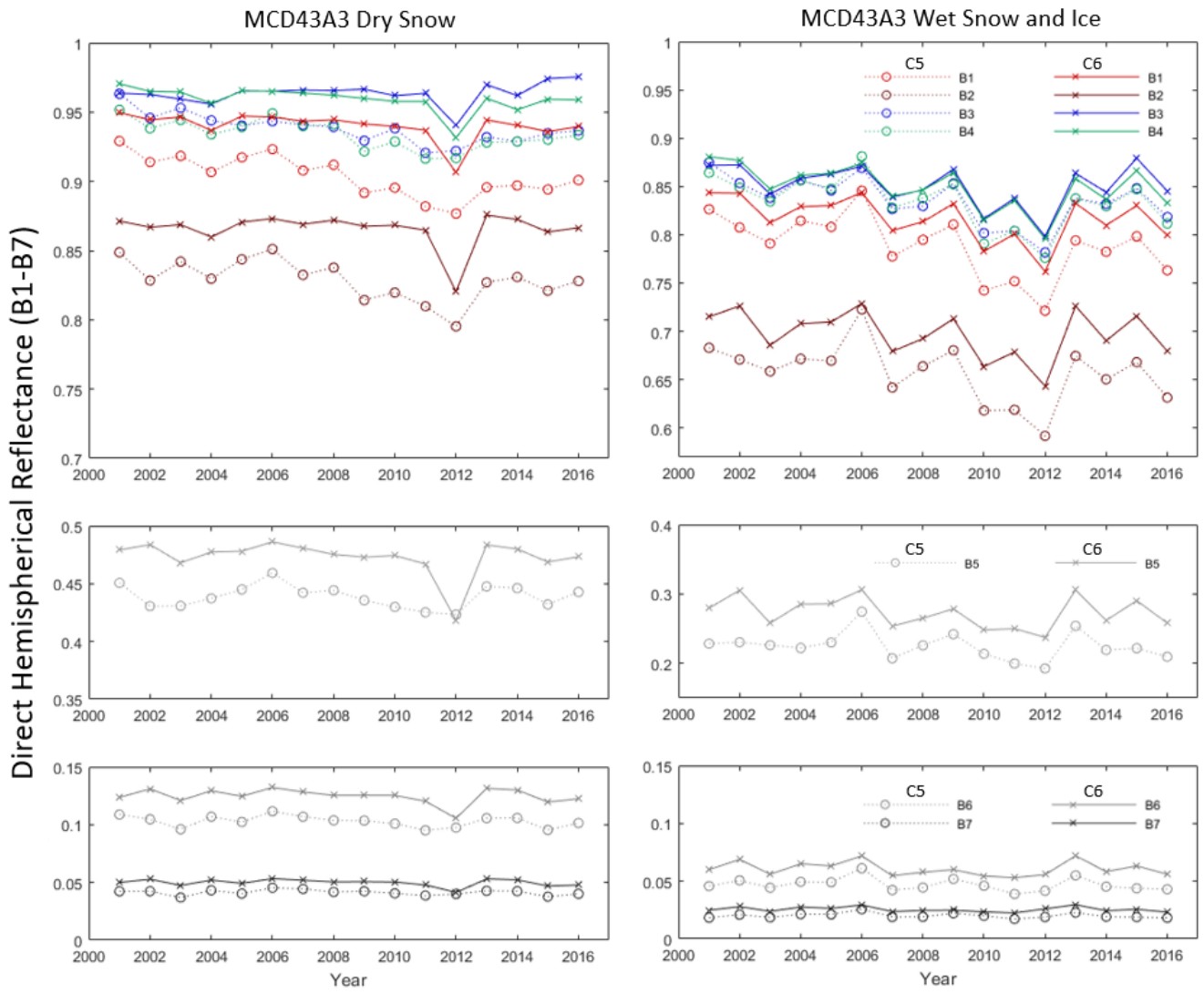

**Figure 4: Greenland ice sheet average summer (Jun 1 - Aug 31) MCD43A3 BSA C5 and C6 Band 1-7 albedo for dry (left) and wet snow and ice (right). (Note the y-axis scale is different for dry vs. wet snow and ice for B1-4 and B5). Interestingly, there is a separation in C5 to C6 B1, B2, B4, B5, B6 albedo values throughout the MODIS record.**

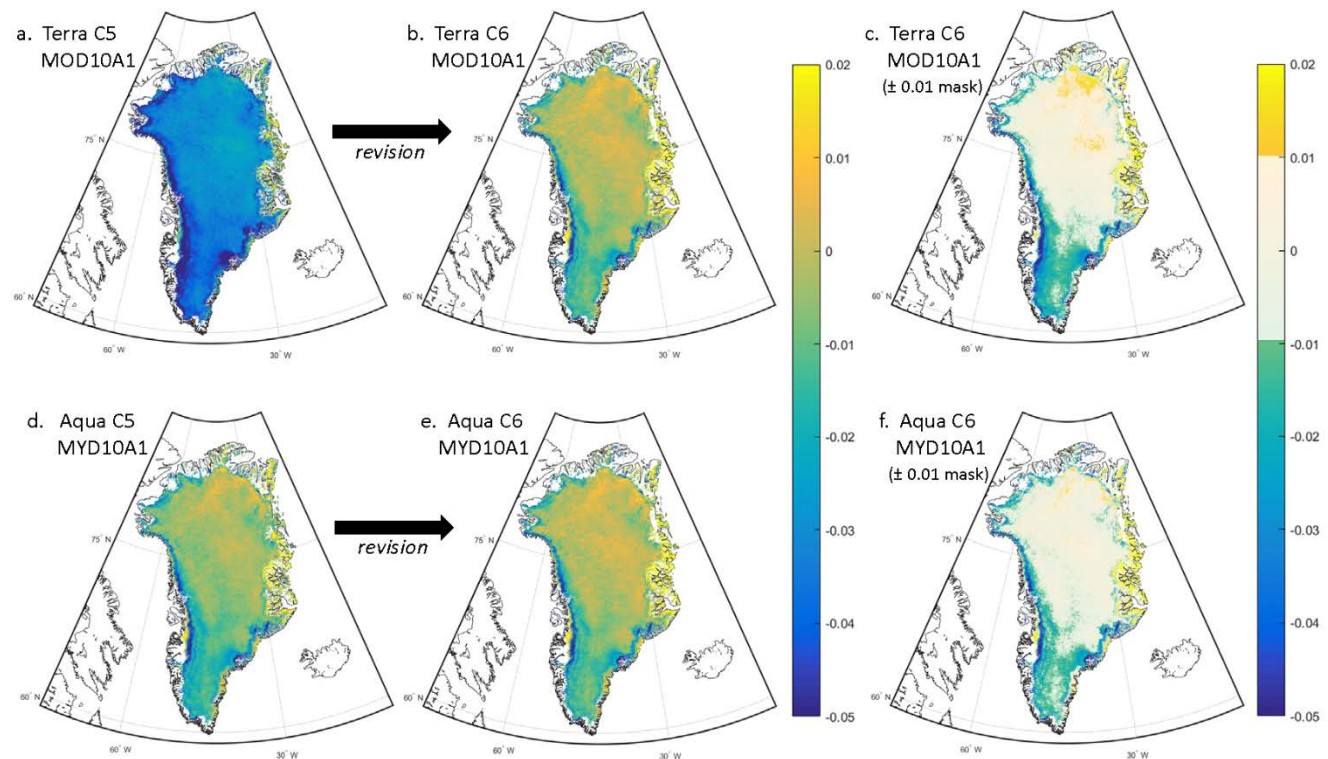

**Figure 5: Maps (a, b) depict Terra MOD10A1 C5 and C6 2002-2016 decadal trend in broadband albedo. Maps (d, e) depict Aqua MYD10A1 C5 and C6 2003-2016 decadal trend in broadband albedo. Maps (c, f) show the same C6 decadal Terra and Aqua trends, respectively, with the sensor trends of ±0.01/decade masked out as white to visualize trends in albedo that are larger than expected**
10  **sensor calibration uncertainty, including albedo decline (southeast and periphery GrIS) and albedo increase (northeast GrIS).**

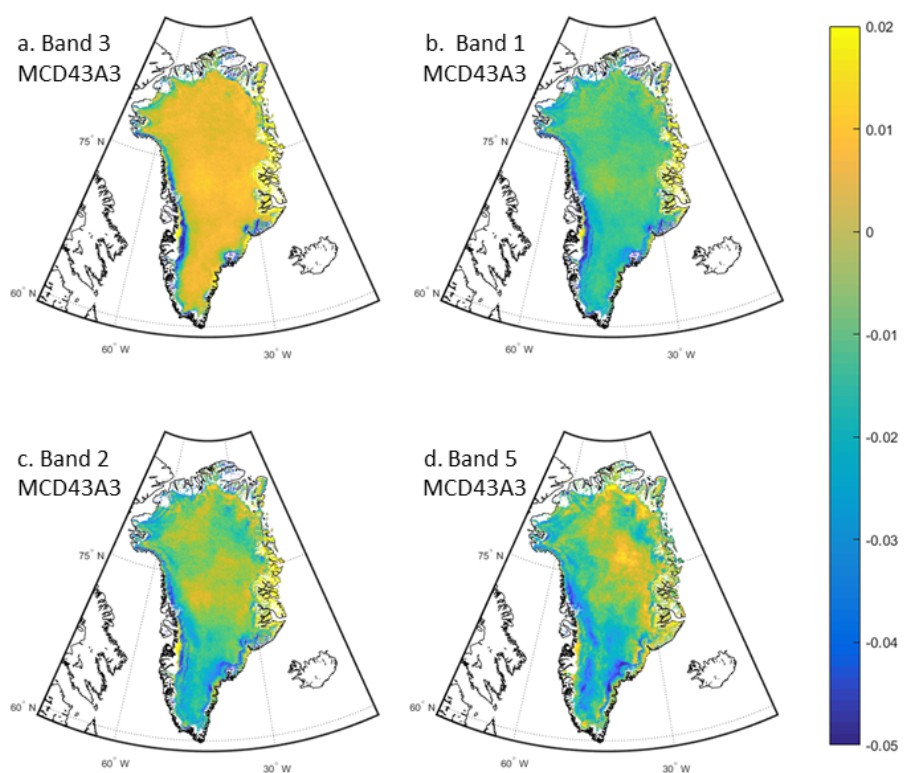

**Figure 6: MCD43A3 C6 band 3 (459-479 nm), band 1 (620-670 nm), band 2 (841-876 nm) and band 5 (1230-1250 nm) 2003-2016 albedo trends. Band-specific maps show considerable spatial complexity in albedo trends.**

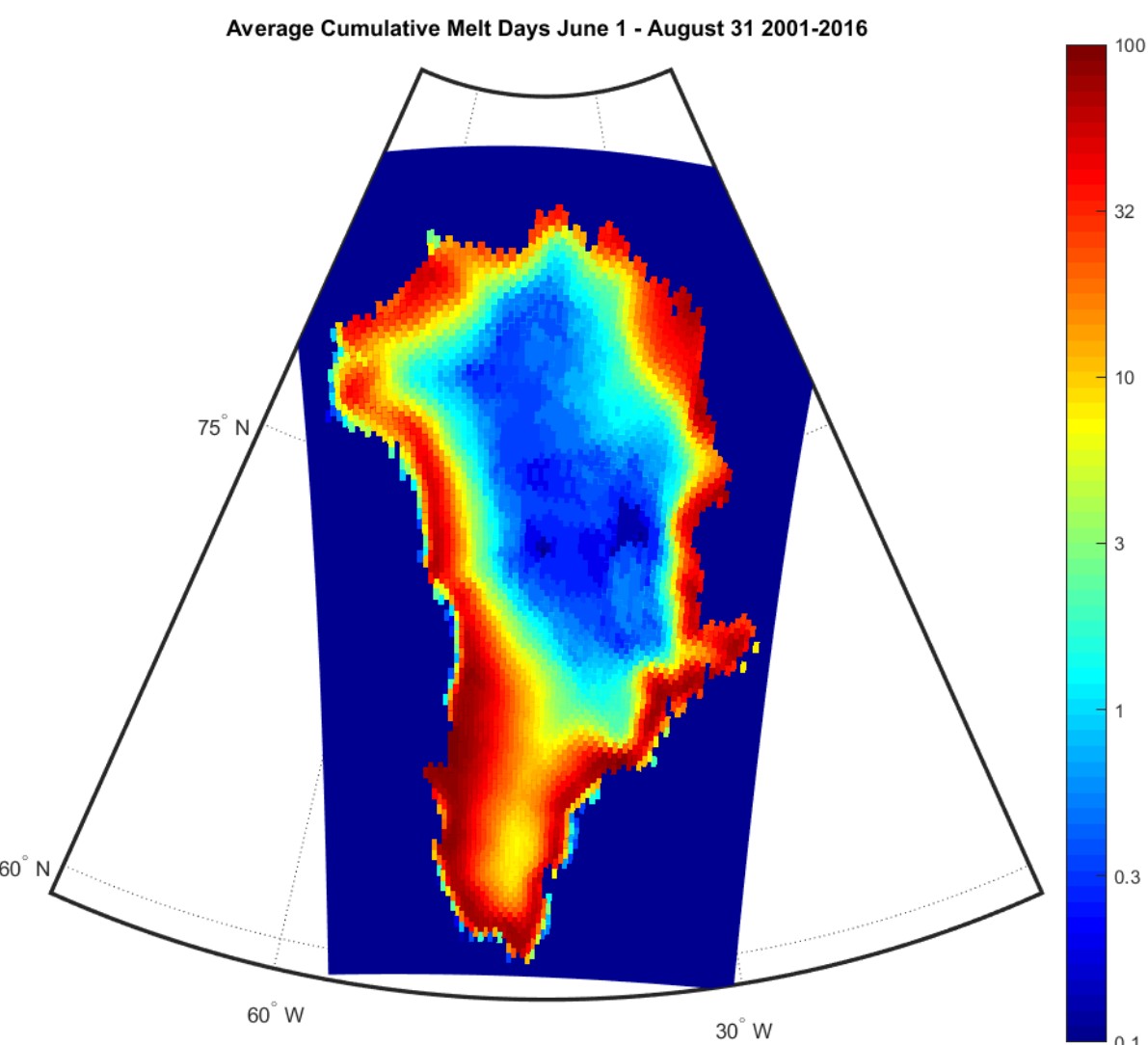

**Figure 7: Average number of melt days experienced from June 1 – Aug 31 as determined from spaceborne passive microwave data [Tedesco, 2014]. Note that the NE GrIS high elevation areas show only a few melt days, almost entirely from 2012. The pattern of melt is inconsistent with the pattern of NIR albedo change. Both low and high elevations in central and NE Greenland show NIR albedo increases while only low elevations experience melt (and the duration is in fact increasing, not shown). Also, the boundary between NIR albedo positive and negative trends falls in the middle of the area with little melt, but closely tracks the summit ridge of the ice sheet, suggesting it is accumulation related rather than melt related.**

**Table 1. MODIS sensor reflective band 1-7 characteristics.**

| MODIS Band | Bandwidth (nm) |
|---|---|
| Band 1 | 620 - 670 |
| Band 2 | 841 - 876 |
| Band 3 | 459 – 479 |
| Band 4 | 545 – 565 |
| Band 5 | 1230 – 1250 |
| Band 6 | 1628 – 1652 |
| Band 7 | 2105 – 2155 |

**Table 2.** Trends (per decade) and statistical significance of trends for M*D09A1 surface reflectance bands 1-7 and M*D10A1 broadband albedo. Statistically significant (where p ≤ 0.05) trends are in bold, and marginally significant trends (where p = 0.05 to 0.1) are in italics.

| Band | Terra C5 Dry Snow Trend | Terra C5 Dry Snow Significance | Terra C6 Dry Snow Trend | Terra C6 Dry Snow Significance | Aqua C5 Dry Snow Trend | Aqua C5 Dry Snow Significance | Aqua C6 Dry Snow Trend | Aqua C6 Dry Snow Significance |
|---|---|---|---|---|---|---|---|---|
| B1 | **-0.027** | **0.003** | -0.007 | 0.166 | -0.006 | 0.377 | -0.002 | 0.761 |
| B2 | **-0.027** | **0.017** | -0.004 | 0.641 | -0.010 | 0.289 | -0.002 | 0.815 |
| B3 | **-0.051** | **0.000** | 0.003 | 0.493 | -0.001 | 0.902 | 0.004 | 0.509 |
| B4 | **-0.018** | **0.013** | *-0.009* | *0.073* | -0.002 | 0.718 | -0.001 | 0.797 |
| B5 | -0.009 | 0.326 | -0.005 | 0.600 | -0.012 | 0.221 | -0.012 | 0.232 |
| B6 | -0.008 | 0.166 | -0.006 | 0.166 | -0.007 | 0.211 | -0.007 | 0.227 |
| B7 | -0.004 | 0.301 | -0.004 | 0.162 | -0.004 | 0.263 | -0.004 | 0.292 |
| Broadband Albedo | **-0.032** | **0.003** | -0.002 | 0.808 | -0.011 | 0.231 | -0.008 | 0.419 |

| Band | Terra C5 Wet Snow/Ice Trend | Terra C5 Wet Snow/Ice Significance | Terra C6 Wet Snow/Ice Trend | Terra C6 Wet Snow/Ice Significance | Aqua C5 Wet Snow/Ice Trend | Aqua C5 Wet Snow/Ice Significance | Aqua C6 Wet Snow/Ice Trend | Aqua C6 Wet Snow/Ice Significance |
|---|---|---|---|---|---|---|---|---|
| B1 | **-0.037** | **0.029** | -0.023 | 0.102 | -0.023 | 0.198 | -0.020 | 0.222 |
| B2 | **-0.042** | **0.024** | -0.020 | 0.194 | -0.026 | 0.166 | -0.022 | 0.223 |
| B3 | **-0.054** | **0.003** | -0.017 | 0.227 | -0.019 | 0.259 | -0.018 | 0.281 |
| B4 | *-0.029* | *0.069* | *-0.025* | *0.073* | -0.020 | 0.237 | -0.020 | 0.202 |
| B5 | **-0.027** | **0.043** | -0.014 | 0.257 | -0.023 | 0.144 | -0.024 | 0.115 |
| B6 | **-0.014** | **0.001** | *-0.006* | *0.067* | -0.008 | 0.102 | *-0.009* | *0.087* |
| B7 | **-0.008** | **0.001** | *-0.003* | *0.075* | -0.004 | 0.123 | -0.005 | 0.109 |
| Broadband Albedo | **-0.040** | **0.001** | -0.013 | 0.166 | -0.009 | 0.364 | -0.007 | 0.495 |

**Table 3. Trends (per decade) and statistical significance of trends for MCD43A3 directional hemispherical reflectance (or black-sky albedo, BSA) and bihemispherical reflectance (or white-sky albedo, WSA). Statistically significant (where p ≤ 0.05) trends are in bold, and marginally significant trends (where p = 0.05 to 0.1) are in italics.**

| | BSA C5 Dry Snow | | BSA C6 Dry Snow | | WSA C5 Dry Snow | | WSA C6 Dry Snow | |
|---|---|---|---|---|---|---|---|---|
| **Band** | **Trend** | **Significance** | **Trend** | **Significance** | **Trend** | **Significance** | **Trend** | **Significance** |
| **B1** | **-0.023** | **0.001** | *-0.009* | *0.096* | **-0.023** | **0.001** | *-0.009* | *0.093* |
| **B2** | **-0.017** | **0.026** | -0.005 | 0.499 | **-0.017** | **0.030** | -0.005 | 0.517 |
| **B3** | **-0.017** | **0.001** | 0.004 | 0.343 | **-0.017** | **0.001** | 0.004 | 0.334 |
| **B4** | **-0.014** | **0.006** | **-0.010** | **0.037** | **-0.014** | **0.006** | **-0.010** | **0.034** |
| **B5** | -0.003 | 0.596 | -0.009 | 0.288 | -0.003 | 0.652 | -0.009 | 0.306 |
| **B6** | -0.004 | 0.141 | -0.004 | 0.317 | -0.004 | 0.150 | -0.003 | 0.345 |
| **B7** | -0.001 | 0.344 | -0.002 | 0.252 | -0.001 | 0.376 | -0.002 | 0.263 |

| | BSA C5 Wet Snow/Ice | | BSA C6 Wet Snow/Ice | | WSA C5 Wet Snow/Ice | | WSA C6 Wet Snow/Ice | |
|---|---|---|---|---|---|---|---|---|
| **Band** | **Trend** | **Significance** | **Trend** | **Significance** | **Trend** | **Significance** | **Trend** | **Significance** |
| **B1** | **-0.0369** | **0.0286** | *-0.0230* | *0.0645* | **-0.0357** | **0.0293** | *-0.0230* | *0.0644* |
| **B2** | *-0.0293* | *0.0842* | -0.0176 | 0.1979 | -0.0269 | 0.1009 | -0.0175 | 0.2004 |
| **B3** | **-0.0292** | **0.0266** | -0.0133 | 0.2745 | **-0.0282** | **0.0289** | -0.0133 | 0.2742 |
| **B4** | **-0.0296** | **0.0412** | **-0.0254** | **0.0376** | **-0.0292** | **0.0397** | **-0.0255** | **0.0373** |
| **B5** | -0.0115 | 0.3073 | -0.0135 | 0.2765 | -0.0106 | 0.3469 | -0.0136 | 0.2741 |
| **B6** | -0.0031 | 0.3205 | -0.0028 | 0.4178 | -0.0030 | 0.3568 | -0.0029 | 0.4038 |
| **B7** | -0.0009 | 0.4445 | -0.0010 | 0.4103 | -0.0008 | 0.5199 | -0.0010 | 0.3958 |