# Peer review of "Impact of MODIS sensor calibration updates on Greenland ice sheet surface reflectance and albedo trends"

_The Cryosphere, 2017_

## Referee Comment (RC1) · JEB Box (Referee) · 1 Apr 2017

The study presents a large multi data set intercomparison, is clearly written and is a valuable contribution to the study of snow (and ice) albedo. The WMO in 2011 has defined snow as an Essential Climate Variables for monitoring.

+ World Meteorlorgical Organization (WMO) 2011 Systematic observation requirements for satellite-based data products for climate. GCOS-154, 138 pp.

The "beyond the scope of this study" argument to not validate using e.g. ground observations is not that convincing. A better justification may be something like: 'While we do not not compare Collection 5 and 6 with ground data, our validation is through

intercomparison of different satellite Terra and Aqua MOD09 and MOD10 products. Further, in an accepted article, Box et al. (2017) find Terra MOD10A1 albedo improving in relative accuracy substantially from Collection 5 to Collection 6, agreeing with GC-Net and PROMICE station data within 0.04 especially in mid-summer and for the majority of the island south of 80 N latitude (Box et al. 2017, Fig 4b).

+ Box, J.E., D. van As, K. Steffen, Greenland, Canadian and Icelandic land ice albedo grids (2000-2016), Geological Survey of Denmark and Greenland Bulletin, Vol. 38, 2017 https://www.dropbox.com/s/4k36mfvackl8m1n/Box%20et%20al.%202017%20-%20revised%20-%20post-review.pdf?dl=0

The definition of wet and dry surfaces is adequate. Yet, reminding the reader throughout the paper that much of the trends in the wet area are for ice not snow, seems important. For example, in the abstract "Wet snow albedo decline ..." I think this surface has a large bare ice fraction. So, something like "Wet snow and ice albedo decline" is more accurate.

In abstract, to be less qualitative, pg 1 line 21 include a quantitative metric beside 'slightly detected' also beside 'lower magnitude' in the next line.

p 7 line 4 'negligible trends' averaging over the sunlit year but what about July when metamorphism and impurity concentration on surface may be strongest?

p 8 line 31 remove 'significant' as the term should be reserved for statistical tests. Here, the use is ambiguous.

To orient readers, all geographic locations referred to in text, e.g. Humboldt should appear on at least one of the maps.

I suggest use of 'area' instead of inconsistent use of 'zone' 'area' and 'region'.

---

## Referee Comment (RC2) · JEB Box (Referee) · 4 Apr 2017

I mentioned a post-review paper in my review and now have a complete citation...

Box, J.E., D. van As, K. Steffen, 2017. Greenland, Canadian and Icelandic land ice albedo grids (2000-2016), Geological Survey of Denmark and Greenland Bulletin, 38, 69-72.

---

## Referee Comment (RC3) · Anonymous Referee #2 · 24 Apr 2017

General comments:

The paper carefully examines how MODIS sensor calibration impacts estimates of surface reflectance and surface albedo on the Greenland ice sheet (GrIS). The issue is that results from the old Collection 5 (C5) showed a clear decreasing trend in GrIS albedo from Terra data. In this paper, newly calibrated results show that the differences between the old C5, which suffers from the problem of calibration drift, and the new C6 are significant. After the correction, results between Terra and Aqua become mutually consistent (Fig. 4), and the erroneous results from C5 are largely removed. This finding is certainly important because misidentifying albedo trend over the Greenland ice sheet may lead to erroneous conclusion on impacts of climate change because albedo

is a key parameter determining surface energy balance. In this regard, utmost care should be taken to ensure the quality of remote sensing data, and I highly commend the authors for making the effort to correct the data calibration problem.

The paper is well written and the figures are well presented (especially Fig. 4). I strongly recommend this paper for publication.

Specific comments:

I do have a number of comments for the authors to consider in improving the paper presentation:

1. Abstract, line 21: "Wet snow albedo" means "Albedo in the wet snow zone"?

2. Abstract, line 24: Can you be more specific about "other fields"?

3. Introduction, line 27: "past two decades" means "1997-2017" or otherwise? Some references you quoted was back to 2000?

4. In the introduction, explain how this paper advances beyond the results by Polashenski et al. (2015) so readers can see right away the significance of the results presented in subsequent sections of this paper.

5. Page 3, line 6: How stable is "stable reference over time" for how long the time is?

6. Page 3, line 25: Any bias expected in the reflectance and albedo products due to residual cloud shadow effects between melt and non-melt zones? Some reference should be good.

7. Page 4, line 22: 0.04 over what value range?

8. Page 4, line 29: "dry and wet snow regions" means "dry and wet snow zones" or otherwise? Check for wording consistency.

9. Include in a relevant section of the paper a map of Greenland DEM contours and names of places over the GrIS.

10. Section 4.3 Spatial Pattern of Albedo Trend: As seen in both Aqua and Terra after correction, Figs. 4 c and f hint that the overall pattern in southern Greenland may have some correlation/consistency with topography (thus good to include a DEM contour map as suggested in item 4), e.g., areas of Saddle and South Dome. What is the main factor causing the albedo change in southern Greenland, melt/snow grain change or back carbon/dust change, and why such factor is related to or governed by topography? Some discussion on this observation may be interesting to include, and perhaps can be related the trend of snow/ice change in southern Greenland influenced or modulated by topography.

11. Section 5.1, page 7, lines 27-28: "decadal trend of declining reflectance at a rate of up to several percent per decade" is significant. It is interesting to highlight the significant by equating such change to an equivalent of how much heat absorption trend (i.e., how much it impacts surface heat balance). This is only a suggestion and the authors can decide to include it or not.

12. Page 8, lines 14-15: Good if you can verify the "enhanced snowfall in the dry regions of NE Greenland." Simply provide a reference if such fact has been published. Similarly, verify "increasing grain size on southern and western areas of GrIS" and provide a reference if available.

13. Section 5.2, Item 2: Good that you point out "degradations can by magnified in band ratio products." Any expected impacts of such degradations on some specific MODIS products that you can point out as examples?

14. Future work: What is the plan to furthermore verify C6 Terra/Aqua results of reflectance/albedo by comparing remote sensing results with independent field observations to assure the quality of both?

---

## Author Comment (AC1) · 16 Jun 2017

**TCD Responses: Impact of MODIS sensor calibration updates on Greenland ice sheet surface reflectance and albedo trends**

*Thank you Dr. Marie Dumont for serving as an editor for this manuscript. Please find our responses in red bold italics below to reviewer comments (in black). We also thank Referee #1, Dr. Jason Box and Anonymous Referee #2 for their constructive comments, which have strengthened this manuscript.*

**Response to Referee #1, Dr. Jason Box**

**Referee #1 comments:**

The study presents a large multi data set intercomparison, is clearly written and is a valuable contribution to the study of snow (and ice) albedo. The WMO in 2011 has defined snow as an Essential Climate Variables for monitoring.

+ World Meteolorgical Organization (WMO) 2011 Systematic observation requirements for satellite-based data products for climate. GCOS-154, 138 pp.

The "beyond the scope of this study" argument to not validate using e.g. ground observations is not that convincing. A better justification may be something like: 'While we do not not compare Collection 5 and 6 with ground data, our validation is through intercomparison of different satellite Terra and Aqua MOD09 and MOD10 products. Further, in an accepted article, Box et al. (2017) find Terra MOD10A1 albedo improving in relative accuracy substantially from Collection 5 to Collection 6, agreeing with GC-Net and PROMICE station data within 0.04 especially in mid-summer and for the majority of the island south of 80 N latitude (Box et al. 2017, Fig 4b).

+ Box, J.E., D. van As, K. Steffen, Greenland, Canadian and Icelandic land ice albedo grids (2000-2016), Geological Survey of Denmark and Greenland Bulletin, Vol. 38, 2017
https://www.dropbox.com/s/4k36mfvackl8m1n/Box%20et%20al.%202017%20-%20revised%20-%20post-review.pdf?dl=0

*Thank you Dr. Box for your careful review of our article and constructive feedback. We thank you for the references, and alerting us to the highly relevant study, which includes recent in situ data comparison. We have edited the manuscript to include the study analyzing Greenland ice sheet ground data with MODIS collection 5 and 6 data as recommended. We have also included other relevant in situ studies.*

The definition of wet and dry surfaces is adequate. Yet, reminding the reader throughout the paper that much of the trends in the wet area are for ice not snow, seems important. For example, in the abstract "Wet snow albedo decline ..." I think this surface has a large bare ice fraction. So, something like "Wet snow and ice albedo decline" is more accurate.

*We thank you for the suggestion and agree. We have updated the description of the Greenland ice sheet surface throughout the manuscript to include appropriate descriptions of snow and ice surface types.*

In abstract, to be less qualitative, pg 1 line 21 include a quantitative metric beside 'slightly detected' also beside 'lower magnitude' in the next line.

*We find it would be too cumbersome to add a metric for each product band per sensor here. We address specific metrics in the tables, figures, results and discussion in the body of the manuscript.*

p 7 line 4 'negligible trends' averaging over the sunlit year but what about July when metamorphism and impurity concentration on surface may be strongest?

*We agree that the change in surface reflectance is often largest from early spring to mid-summer. We agree that for many years of the MODIS record, Greenland ablation may be strongest in July when snow melts and reveals bare ice, and/or wet snow and impurities concentrate at the surface. As mentioned in section 3, we previously completed the three MODIS product comparison for the time period May 15 – July 15, finding very similar results. We expect that a July 1 – July 31 comparison would yield similar overall results and revisions to C5 vs. C6 trends.*

p 8 line 31 remove 'significant' as the term should be reserved for statistical tests. Here, the use is ambiguous.

*The sentence has been reworded to avoid use of 'significant' as recommended.*

To orient readers, all geographic locations referred to in text, e.g. Humboldt should appear on at least one of the maps.

*As recommended, we have added a figure (Figure 1) detailing the surface elevation of the Greenland ice sheet, with 500m, 1000m, 2000m and 3000m elevation contour lines as well as names of places referred to in the manuscript.*

I suggest use of 'area' instead of inconsistent use of 'zone' 'area' and 'region'.
*We have edited the text as recommended.*

I mentioned a post-review paper in my review and now have a complete citation...

Box, J.E., D. van As, K. Steffen, 2017. Greenland, Canadian and Icelandic land ice albedo grids (2000-2016), Geological Survey of Denmark and Greenland Bulletin, 38, 69-72.

*Thank you for the notice; we have added this citation.*

**Response to Anonymous Referee #2:**

General comments: The paper carefully examines how MODIS sensor calibration impacts estimates of surface reflectance and surface albedo on the Greenland ice sheet (GrIS). The issue is that results from the old Collection 5 (C5) showed a clear decreasing trend in GrIS albedo from Terra data. In this paper, newly calibrated results show that the differences between the old C5, which suffers from the problem of calibration drift, and the new C6 are significant. After the correction, results between Terra and Aqua become mutually consistent (Fig. 4), and the erroneous results from C5 are largely removed. This finding is certainly important because misidentifying albedo trend over the Greenland ice sheet may lead to erroneous conclusion on impacts of climate change because albedo is a key parameter determining surface energy balance. In this regard, utmost care should be taken to ensure the quality of remote sensing data, and I highly commend the authors for making the effort to correct the data calibration problem.

The paper is well written and the figures are well presented (especially Fig. 4). I strongly recommend this paper for publication.

*We thank the Anonymous Reviewer for his or her careful review of the manuscript and constructive suggestions for further improvement.*

Specific comments: I do have a number of comments for the authors to consider in improving the paper presentation:

1. Abstract, line 21: "Wet snow albedo" means "Albedo in the wet snow zone"?

*Correct, we have rewritten this sentence for clarity.*

2. Abstract, line 24: Can you be more specific about "other fields"?

*We have edited the sentence to clarify that investigators of other ocean, atmosphere and/or land analyses are urged to consider re-evaluating MODIS trends.*

3. Introduction, line 27: "past two decades" means "1997-2017" or otherwise? Some references you quoted was back to 2000?

*Thank you for pointing out this oversight. We have modified this sentence to specify the record of 30 years of mass loss based on published literature.*

4. In the introduction, explain how this paper advances beyond the results by Polashenski et al. (2015) so readers can see right away the significance of the results presented in subsequent sections of this paper.

*This paper advances beyond the results of Polashenski et al. (2015) by completing the MODIS Collection 6 three data product analysis over the full MODIS record, adding more spectral bands and investigating spatial change over time by broadband and individual spectral bands. The paper also describes spatial surface reflectance and albedo patterns observed in the new Collection 6 data as compared to old Collection 5 data, pointing out sensor degradation influences. We suggest that the spectral and spatial patterns identified in our satellite-based data study be further investigated by field, modeling and remote sensing studies. We have rewritten parts of the introduction as recommended to clarify this as recommended.*

5. Page 3, line 6: How stable is "stable reference over time" for how long the time is?

*As referenced, Toller et al., 2013 provides detail on MODIS instrument Collection 6 calibration and L1B data product software, which issues scan collected corrections and data quality/stability metrics. The MODIS Characterization Support Team (https://mcst.gsfc.nasa.gov) provides up to date information on MODIS instrument characteristics and performance, including assessment of instrument stability over time. We have cautioned that it will take time to understand how well C6 updates address the sensor degradations. The targeted sentence was modified.*

*Toller, G., X. Xiong, J. Sun, B.N. Wenny, X. Geng, J. Kuyper, A. Angal, H. Chen, S. Madhavan, A. Wu, (2013), Terra and Aqua moderate-resolution imaging spectroradiometer collection 6 level 1B algorithm, Journal of Applied Remote Sensing, 7, 1, doi:10.1117/1.JRS.7.073557.*

6. Page 3, line 25: Any bias expected in the reflectance and albedo products due to residual cloud shadow effects between melt and non-melt zones? Some reference should be good.

*There is a potential impact in the MODIS reflectance and albedo products due to residual cloud shadow effects. We have added references related to changes in C6 cloud and aerosol processing algorithms.*

7. Page 4, line 22: 0.04 over what value range?

*In the M\*D10A1 processing, albedo ranges from 0.0-1.0, pixels were removed which were more than 2 standard deviations from the 11-day running median or which differ from the 11-day running median by more than 0.04 (on a scale of 0.0-1.0) after Box et al., 2012. We reworded the sentence to be clear.*

8. Page 4, line 29: "dry and wet snow regions" means "dry and wet snow zones" or otherwise? Check for wording consistency.

*We have edited the dry and wet snow and ice areas description throughout the manuscript to be more consistent.*

9. Include in a relevant section of the paper a map of Greenland DEM contours and names of places over the GrIS.

*We have added Figure 1 to show the elevation of the Greenland ice sheet, with 500m, 1000m, 2000m and 3000m elevation contour lines as well as names of places referred to in manuscript.*

10. Section 4.3 Spatial Pattern of Albedo Trend: As seen in both Aqua and Terra after correction, Figs. 4 c and f hint that the overall pattern in southern Greenland may have some correlation/consistency with topography (thus good to include a DEM contour map as suggested in item 4), e.g., areas of Saddle and South Dome. What is the main factor causing the albedo change in southern Greenland, melt/snow grain change or back carbon/dust change, and why such factor is related to or governed by topography? Some discussion on this observation may be interesting to include, and perhaps can be related the trend of snow/ice change in southern Greenland influenced or modulated by topography.

*We thank you for the comment. We have added Figure 1 with the topographic information that can be studied with reported spectral and spatial declining reflectance and albedo patterns and trends. Recent studies have investigated orographic effects on snow deposition (e.g. impurity deposition of soot-laden air masses) and increased flow/velocity with steeper slopes and/or more lubricated basal conditions. The reviewer is correct to suggest further study be conducted to determine the exact factors causing albedo change in southern Greenland, including in situ black carbon, dust and snow grain studies as well as snow and ice physical processes.*

11. Section 5.1, page 7, lines 27-28: "decadal trend of declining reflectance at a rate of up to several percent per decade" is significant. It is interesting to highlight the significant by equating such change to an equivalent of how much heat absorption trend (i.e., how much it impacts surface heat balance). This is only a suggestion and the authors can decide to include it or not.

*The quoted phrase "decadal trend of declining reflectance at a rate of up to several percent per decade" refers to C5 data dry snow area findings. MODIS C6 dry snow area findings show declining reflectance, though apart from MCD43 B4, are not statistically significant.*

*In the wet snow areas, MODIS C6 MCD43 data shows statistically significant decline in albedo. We have added references to other studies equating similar declines in reflectance to impacts of radiative energy balance, namely Painter et al., 2007; Casey et al. 2017.*

12. Page 8, lines 14-15: Good if you can verify the "enhanced snowfall in the dry regions of NE Greenland." Simply provide a reference if such fact has been published. Similarly, verify "increasing grain size on southern and western areas of GrIS" and provide a reference if available.

*Thank you for the suggestion. Unfortunately, data that could support or disprove these trends is sparse at best. Our intent is to state what the spectral signature seems to indicate as a possible physical process to motivate further study. We explicitly state the inference is speculative. Later in the section, we detail the published literature that suggests increased snowfall has been observed and*

*is found in model results in NE Greenland.   Specifically, the last paragraph in Section 5.1, we detail that MAR data indicate increased snowfall in the dry regions of NE Greenland since 2013.  We are unable to find data that substantially supports or contradicts the indication of grain size growth in southern dry snow area, beyond satellite speculation and anecdotal evidence.*

13. Section 5.2, Item 2: Good that you point out "degradations can by magnified in band ratio products." Any expected impacts of such degradations on some specific MODIS products that you can point out as examples?

*We thank the reviewer for this comment. The authors are not familiar with the algorithms of every MODIS data product nor MODIS study.  Therefore, we noted the potential impacts to dust mineralogy and aerosols, known science application areas utilizing the MODIS early visible bands (e.g. B3, B4, B8) which have traditionally experienced the greatest calibration degradation (Toller et al., 2013).*

14. Future work: What is the plan to furthermore verify C6 Terra/Aqua results of reflectance/albedo by comparing remote sensing results with independent field observations to assure the quality of both?

*We have recommended at the end of section 5.1 further suggested in situ studies to test the hypotheses formed from the MODIS data product analysis.  The grants that originally supported this work have ended.*

[revised manuscript text omitted]